



# Maximum summer temperatures predict the temperature adaptation of Arctic soil bacterial communities

Ruud Rijkers[1], Mark Dekker[1], Rien Aerts[1] and James T. Weedon[1]

Amsterdam Institute for Life and Environment, Section of Systems Ecology, Vrije Universiteit Amsterdam, Netherlands

*Correspondence to:* Ruud Rijkers (r.rijkers@vu.nl)

**Abstract**

Rapid warming of the arctic terrestrial region has the potential to increase soil decomposition rates and form a carbon-driven
feedback to future climate change. For accurate prediction of the role of soil microbes in these processes it will be important
to understand the temperature responses of soil bacterial communities and implement them into biogeochemical models. The
temperature adaptation of soil bacterial communities for a large part of the Arctic region is unknown. We evaluated the current
temperature adaption of soil bacterial communities from 12 sampling sites in the sub- to High Arctic. Temperature adaptation
differed substantially between the soil bacterial communities of these sites, with estimates of optimal growth temperature
(Topt) ranging between 23.4 ± 0.5 and 34.1 ± 3.7 C. We evaluated possible statistical models for the prediction of the
temperature adaption of soil bacterial communities based on different climate indices derived from soil temperature records,
or on bacterial community composition data. We found that highest daily average soil temperature was the best predictor for
the Topt of the soil bacterial communities, increasing 0.63 °C per °C. We found no support for the prediction of temperature
adaptation by regression tree analysis based on relative abundance data of most common bacterial species. Increasing summer
temperatures will likely increase Topt of soil bacterial communities in the Arctic. Incorporating this mechanism into soil
biogeochemical models and combining it with projections of soil temperature will help to reduce uncertainty in assessments
of the vulnerability of soil carbon stocks in the Arctic.

## 1.   Introduction

The Arctic terrestrial biome has the potential to undergo particularly large losses of soil organic carbon and controls the
potential loss or gain of global carbon stocks (Crowther et al., 2016; Wieder et al., 2019). This is because of the large soil
organic carbon stock in arctic soils (Tarnocai et al., 2009) and the strong  response of soil respiration rates to warming in these
cold ecosystems (Carey et al., 2016). Bacterial soil communities in the Arctic terrestrial region are adapted to perform well at





low temperatures (Bååth, 2018). However, these bacterial communities are likely to be exposed to increasing soil temperatures in this century (Post et al 2018) and it remains uncertain whether these soil bacterial communities will adapt their response
to temperature when exposed to warmed conditions (Rinnan et al., 2011; Weedon et al., 2022; Rousk et al., 2012). Knowledge of the climate conditions under which such an adaption takes place will help in estimations of the potential vulnerability of arctic soil carbon stocks to warmer climate conditions (Bååth, 2018; Bradford et al., 2019; Palacios et al., 2021).

The temperature adaptation of soil bacterial communities is most often characterized in relation to respiration, growth or enzymatic activity. A commonly applied method is to estimate the relationship between whole community growth and
temperature with an assay that measures $^3$H-leucine uptake (Bååth et al., 2001). This relationship between temperature and bacterial growth can be described by the Ratkowsky model, which has three cardinal points: the (theoretical) minimum growth temperature (Tmin), optimal growth temperature (Topt) and maximum growth temperature (Tmax) (Ratkowsky et al., 1983). Previous research has shown that the temperature-growth relationships of soil bacterial communities adapt to their local environment, such that there is a positive correlation between mean annual air temperature (MAAT) and the parameters
describing the temperature-growth relationships of soil bacterial communities (cardinal points) (Bååth, 2018). For example, recently it has been found that across an elevation gradient in the Peruvian Andes Tmin increased 0.2 degrees per degree Celsius increase in MAAT (Nottingham et al., 2019) and a similar correlation was found between MAAT and Topt across a natural climate gradient in Europe (Cruz Paredes et al., 2021). This correlation has also been shown in the Antarctic, where the temperature-growth relationships of soil bacterial communities show higher values of Tmin with higher mean annual soil
temperature (Rinnan et al., 2009). However, no comparable large-scale study on the temperature-growth relationships of soil bacterial communities in the Arctic has been performed yet. Such a large scale study is needed for arctic soil bacterial communities, as the Arctic differs from lower latitudinal regions in terms of its current climate(Convey, 2013), predicted climate changes (Post et al., 2019) and importance for the global soil carbon stock (Wieder et al., 2019).

Despite strong correlations over large spatial scales, an increase in the mean annual soil temperature does not necessarily
lead to a shift in temperature-growth relationships of bacterial communities when soils are experimentally warmed in lab incubation and field studies (Pietikäinen et al., 2005; Birgander et al., 2013, 2018; Rinnan et al., 2011; Weedon et al., 2022). Instead, a common observation is a rapid change in the temperature-growth relationships driven by a community turnover when soils are incubated above the optimal growth temperature of the *in situ* soil bacterial community, (Birgander et al., 2013; Donhauser et al., 2020). This suggests that the *maximum* soil temperature is an
important predictor of the temperature-growth relationships of bacterial communities. Supporting evidence for this comes from a study in the Antarctic, where coastal water bacterial communities are adapted to lower temperatures (lower Tmin) than soil bacterial communities in the same region, despite the mean annual temperature of Antarctic water being higher than that of Antarctic soils (van Gestel et al., 2020). The Antarctic soils are exposed to higher summer temperatures than the Antarctic marine environment, leading to the hypothesis that the maximum temperature, rather



than the annual average, is a more important driver for the temperature adaptation of bacterial communities across different habitats (Birgander et al., 2013; van Gestel et al., 2020).

Analogous to the maximum temperature, the coldest soil temperature could also influence temperature-growth relationships. In desert soils, the upper layer (0-5 cm) is characterized by relatively large amplitude fluctuations in temperature over both diurnal and annual timescales. Consequently, the bacterial communities of these upper layers tend

to have lower Tmin values and higher Topt values than deeper soil layers that are exposed to more moderate and stable soil temperatures (van Gestel et al., 2013). These studies show that while the mean annual temperature might correlate strongly with the cardinal points of the temperature-growth relationships of soil bacterial communities, the temperature adaption might be more directly related to other selective pressures of the thermal regime such as the highest or lowest soil temperature.

To predict future temperature-growth relationships of soil bacterial communities in the Arctic, more knowledge is needed on 1) the current temperature adaptation of soil bacterial communities in the Arctic and 2) the specific mechanisms driving temperature adaptation. Bacterial communities from polar ecosystems are hypothesized to be adapted to low temperatures, shown by a low Tmin (Baath, 2018). For example, sub-Arctic bacterial communities exhibit lower cardinal points of their temperature-growth relationships compared to bacterial communities of temperate ecosystems, with a

Tmin of –9.6 to -7.0 °C and Topt 25 to 30 °C (Cruz-Paredes et al., 2021; Rinnan et al., 2011). It is likely that soil warming will shift the temperature-growth relationships of sub-Arctic soil bacterial communities (Weedon et al., 2022; Rijkers et al., 2022). However, the *in situ* temperature-growth relationships of soil bacterial communities in the mid- to High Arctic are so far unknown and will need to be evaluated to understand the current temperature adaptation of soil bacterial communities and drivers of temperature adaptation under future climate conditions.

It is important to evaluate which soil thermal parameters are the most accurate predictor for soil bacterial communities in the sub- to High Arctic, as this might not be accurately predicted from the mean soil annual temperature alone. In these (sub-) Arctic regions the maximum and minimum daily soil temperatures are only weakly correlated with the mean annual soil temperature, due to the influence of local environmental parameters on the soil climate extremes. For example, winter soil temperatures also vary greatly on the meter-scale in the Arctic, due to the influence of snow cover

on winter microclimate (Karjalainen et al., 2018). On the other hand, summer soil temperature is more closely related to the air temperature, which varies less between (sub-) arctic soils (Fig. 1). Implementing knowledge about these possible drivers of the temperature adaptation of soil bacterial communities at these high northern latitudes will support accurate predictions of soil decomposition of the large carbon stock present in the Arctic under future climates.





Due to the possible influence of multiple soil thermal parameters, accurate prediction of temperature adaptation by soil
bacterial communities will likely require high-resolution soil temperature data. However, soil temperature logger data
are particularly scarce in the Arctic region (Lembrechts et al., 2021), leading to a need for potential alternative predictors
of soil microbial temperature adaptation.  DNA-based bacterial community composition measures have recently been
shown to correlate with shifts in the temperature growth relationships of a soil bacterial communities (Donhauser et al.,
2020; Rijkers et al., 2022; Weedon et al., 2022). More generally, temperature traits differ between members of bacterial
communities from arctic soils (Wang et al., 2021), and specific bacterial OTUs have been associated with warming in
forest soils across North America (Oliverio et al., 2017). The aggregated community response, such as the temperature-
growth relationship, might therefore be predictable using the abundance of specific species that are associated with a
warm or cold adapted community (Hicks et al., 2021). In a pan-arctic survey soil bacterial community showed a large
diversity of species, with 15 common OTUs shared between all soils (Malard et al., 2019). Therefore, potentially there
are bacterial species that could indicate the current temperature adaptation of arctic soil bacterial communities. If so, this
provides opportunities to determine the temperature adaptation of soil bacterial communities in the Arctic where long
term soil temperature logging is absent.

In this study we tested which soil thermal parameters best predicts the cardinal points of the temperature-growth
relationships of bacterial communities from 12 soils collected in the sub- to high Arctic region. We hypothesized that
the highest and lowest daily soil temperatures would be the best predictor of the corresponding cardinal points of the
temperature-growth relationships. We also compared the DNA-based compositional profiles across soil types and
explored whether such compositional data can be used as an alternative predictor for the temperature-growth
relationships of the soil bacterial communities in Arctic soils.



## 2. Methods

### 2.1 Sample collection

In the summers of 2018 and 2020, soil samples were collected from 12 soil types at 9 sites ranging from sub- to High Arctic (Figure 1). The 2018 sampling at Toolik Lake Field station, Svalbard, Abisko and Iceland has been previously described in (Rijkers et al., 2022). In brief, soil cores of 10 cm depth were collected from Toolik Field Station, USA (68°38' N, 149°36' W) at the LTER Heath site, LTER Moist Acidic Tussock and LTER Non-Acidic Tussock; on Svalbard, from the Bjorndalen site (78°13'N, 15°19'E), dominated by *Carex* sp. vegetation;  at the FORHOT site in Iceland (64° 00′N, 21° 11′W), a grassland (*Agrostis capillaris*) and forest site (*Picea sitchensis*) were sampled. Lastly, soil samples were collected from the blanket bog (*Sphagnum* sp.) where the ITEX warming experiment is located, close to the Abisko Research Station in Sweden (68°21'N, 18°49'E).

In 2020 a second sampling campaign collected triplicate soil cores to a depth of 10 cm at sites in Greenland (two sites), Canada, Norway, and Finland. On Disko Island, Greenland soil cores were collected near the AWS-2 logger at Østerlien site of the Greenland Ecosystem monitoring (GEM; 69°15''' N, 53°30'W), which were covered by *Vaccinium* sp. At Kobbefjord, Greenland soil samples with *Empetrum sp.* cover were collected near the SoilEMP logger of GEM (64°08'N 51°22'W). At Inuvik, Canada soil cores were sampled at Inuvik airport bog (68º 18.9342 N, 133º 26.0214 W), which is characterized by low shrubs (Nixon et al 2003). In Finland, samples were collected directly next to the ITEX site in Kilpisjarvi (69.4 N, 20.490E), for which the vegetation cover is dominated by *Vaccinium* and *Empetrum* sp (Ylänne et al 2015). Lastly, soil samples were collected at Petersfjellet in Norway (N69°35.5277' E29°55.1939'), which was covered by *Empetrum nigrum*.

### 2.2 Soil temperature data

Soil temperature records were collected from the involved research stations (at Abisko (Dorrepaal et al., 2004), Svanhovd (BioForsk Svanhovd; http://lmt.bioforsk.no/agrometbase/getweatherdata_new.php?weatherStationId=36), Inuvik (National Resources Canada), Svalbard (Global Terrestrial Network for Permafrost database; http://gtnpdatabase.org/boreholes/view/166), Toolik Lake (Hobbie and Laundre, 2021), FORHOT research site in Iceland (Sigurdsson et al., 2016), Kilpisjarvi ITEX site (unpublished, personal communication Sari Stark), Greenland sites (Green Ecosystem monitoring database; https://data.g-e-m.dk/)). To overcome differences in the time intervals of data collection between sites, we calculated the mean daily temperature for each day that soil temperature records were available (all records >3 years, except for Kilpisjarvi; Table 1). Based on the daily soil temperature records of each soil, we determined the mean annual temperature (MAT), mean warmest day (MaxT), mean coldest day (MinT) based on the annual records for the warmest day, coldest day and mean daily temperature per year.





### 2.3 Soil analysis

After collection, soils were shipped on ice and cooled upon arrival at 4°C. The upper 10 cm was sampled for the following analyses: density of the soils samples was determined by rapid submersion in a water filled cylinder. The water content was calculated based on the weight before and after drying the soil at 70° C for 48 h. The dried samples were ground with a Retsch MM200 ball mill (Retsch, Haan, Germany) for 1 min at 30 rounds per second. A subsample was then ashed at 600ºC for 6 h. The carbon and nitrogen content of ashed and non-ashed subsamples were measured on a Flash EA 1112 (ThermoFisher, Waltham, USA). For the calculation of organic carbon in the soil, the carbon content of the ashed samples was subtracted from the total amount of carbon content. Soil pH was measured by adding 5 g of soil to 25 ml deionized water, after which the slurries were shaken for 1 h at 100 rpm. The soil pH was then measured on a WTW Inolab level2 pH meter (Xylem Analytics, Rye Brook, New York, USA). The slurry was then centrifuged at 200 rpm, for 1 hr and then filtered on 0.45µm nylon filter. The filtered solution was used for the measured of extractable dissolved organic carbon content on a TOC-L CPH/CPN (Shimadzu, Columbia, USA). with NPOC method by manufacturer's protocol. For the soil samples of Svalbard only pH measurements were performed due to limited amounts of soil.

### 2.4 Temperature-growth relationships of soil bacterial communities

For the assessment of the temperature sensitivity of bacterial growth, 1 gram of soil was subsampled for a leucine incorporation assay using methods adjusted from Bååth et al., (2001). Briefly, 20 ml of sterilized deionized water was added to the soil samples and these slurries were vortexed for 2 min at full speed. After 10 min centrifugation at 1000 G, the 1 ml aliquots of the supernatant were suspended in 2 ml screw-top Eppendorf tubes. A 20 µl mixture $^3$H-labeled and unlabeled leucine was added, resulting in a final concentration of 401 nM and 72.5 kBq ml$^{-1}$ in the assay tube. The sample aliquots were incubated at 0, 4, 10, 15, 24.5, 28.5, 33.5 and 40 °C for 24 – 2 hours. Trichloroacetic acid was added to the assay tubes to terminate the leucine incorporation. Washing steps for removal of non-incorporated leucine were followed as described in (Bååth et al., 2001). For scintillation 1 ml Optiphase HiSafe 3 (PerkinElmer, Waltham, Massachusetts, USA) was finally added to the biomass pellet after the washing steps. 3H-activity was measured on a Tricarb2800T (Perkin Elmer), Waltham, USA) with 5 minute measurement for $^3$H. Finally, the leucine incorporation rate, nM leucine 1 h -1 g dry weight soil, was calculated based on $^3$H activity measured.

### 2.5 Bacterial community composition

For the characterization of the soil microbial community, 0.2 grams of soil were subsampled for DNA extraction and amplicon sequencing of the 16S rRNA gene. DNA was extracted by the use of Powersoil kit (Qiagen, Hilden, Germany), following the manufacturer's protocol with elution of the purified DNA into 60 sterile µl Millipore water. Amplicons



were generated by a two-step PCR of the 16S V4 rRNA gene with primers designed by (Caporaso et al., 2012). An initial PCR consisted of 24 cycles with an initial denaturation step of 1 min at 98°, followed by 25 cycles of denaturation for 10 s at 98 °C, annealing for 30 s at 55°C, elongation for 30 s at 72°C, followed by a final extension of 5 min at 72°C. Amplicons were then 50x diluted in σ-purified water and then indexed by a PCR with unique barcode primers for 8 cycles with the same steps as the initial PCR amplification. Purification of the PCR product was done with Ampure XP

beads (Beckman Coulter, Brea, California, USA), following manufacturer's protocol. The indexed PCR products were then sequenced using paired-end Illumina MiSeq runs with V3-2x300 cycle chemistry. In total 1,243,600 sequences were generated for 39 samples (Median sequencing depth; 32,089 sequences per sample). Sequences were truncated at 250 nucleotides on the forward reads and 220 nucleotides on the reverse reads due to deteriorating quality scores for later cycles (average Phred score < 30). Raw sequences are available in the NCBI Sequence Archive, under BioProject

Accession number PRJNA857550. Amplicon sequence variants (ASVs) were generated by dereplication and chimera removal of the truncated sequences using DADA2 allowing a maximum expected error of 2 and 'consensus' chimera removal mode. Phylogenetic distances between the ASVs were estimated using MAFFT alignment (Katoh and Standley, 2013) and Fasttree (Price et al., 2009). Taxonomic classification of the ASVs was performed based on the SILVA v138 database (Yilmaz et al., 2014) using a scikit-learn naive Bayes machine-learning classifier (Bokulich et al., 2018) with

a confidence threshold for limiting taxonomic depth at 70%. ASVs identified as mitochondria or chloroplasts as well as singletons were discarded prior to further statistical analyses.

**2.6 Statistical analyses**

All statistical analyses were performed in R (v4.0.2) (R Core Team, 2020). Soil daily temperature records were filtered for datapoints between 2002 and 2021. Leucine incorporation rates were fitted to a Ratkowsky model for bacterial growth

(Ratkowsky et al., 1983) by the use of R-package 'nls.multistart' (Padfield and Matheson, 2018). The Ratkowsky model is based on the following equation;

Eq. 1

$$\sqrt{Leu} = a(T - Tmin) \times (1 - e^{b(T-Tmax)})$$

where Leu is the rate of leucine incorporation, a is the coefficient below optimal growth temperature, T is the assay

temperature, Tmin is the theoretical minimum growth temperature, b is coefficient above the optimal growth temperature and Tmax is maximum growth temperature. The optimal temperature was determined by numerical interpolation. All figures were made with the 'ggplot2' R-package. To test for the effects of soil thermal parameters on the temperature adaptation of soil bacterial communities, we performed linear regression between the cardinal points of the temperature-growth relationships and minimum (MinT), mean (MAST), and maximal annual soil temperature (MaxT). These linear





regression models tested the relationship between Tmin and minimum soil temperature, Tmax and the maximum soil temperature and Topt with minimum, mean and maximum soil temperature as independent variable. We fitted a linear regression for the relationship between the temperature range (Tmin – Tmax) of the temperature-growth relationships of the soil bacterial communities and amplitude of thermal soil regime (minimum MinT to maximum soil temperature MaxT) with a linear regression model.

Processing the microbial community data was done using the R-package 'phyloseq' (McMurdie and Holmes, 2013). Samples were rarified to depth of 23687 reads. Permutational multivariate analysis of variance (PERMANOVA; Anderson, 2001) was performed on the weighted UniFrac distances (Lozupone and Knight, 2005) of the sample of the 11 sites, excluding the Svalbard due to lack of data, (Suppl. Table 1.) using the mean annual soil temperature, pH, organic carbon content, organic nitrogen content and community Topt as independent variables in the 'vegan' R-package. We determined the common ASVs

by filtering for mean relative abundance above 0.001 % in at least 4 soil types. The relative abundance of the common ASVs was used to predict the Topt of the soil bacterial communities. The relative abundance of these common ASVS was then used to perform a 3 types of regression tree analysis on the Topt of soil bacterial communities using the R-package 'caret' (Kuhn, 2008). Data were randomly split into training (9 soils) and validation (3 soils) dataset, after which a regression tree analysis was performed with 'rpart1SE' function using the control settings (maxdepth=4, minsplit=4, minbucket =2). We also build a

regression tree with cross validation (10 folds, 10 repeats) using the 'rpart' function using the same control settings. Additionally, we used 'Rborist' function with the default setting to calculate a random forest regression tree to predict Tmin based on the relative abundance of common ASVs in the training soils. For direct comparison with regression models, we performed an additional linear regression using Topt as independent variable and MaxT as dependent variable using the 9 soils of the training dataset and 3 soils in the validation dataset. Due to the small datasets that these models were based on, the

random division into training and validation dataset had a strong influence on the computed RMSE (root mean square error) value. Therefore, we trained each of the 4 models on all 220 possible combinations of soils in the training and validation dataset (with a 9:3 split between soil for train and testing, respectively). We then compared the performance of the 4 different models based on median RMSE over the 220 simulations.



## 3. Results

### 3.1 Temperature adaptation of soil bacterial communities

From sub- to High Arctic, mean annual soil temperatures at 10 cm depth varied between -3.5 and 6.1 °C (Table 1, Figure 1). The sampled bacterial communities varied in Tmin between -11.1 ± 4 (s.d) in Østerlien and -5.5 ± 2.1 in the Icelandic grassland. Topt varied between 23.4± 0.5 in Toolik Lake MAT and 34.1 ± 3.7 in Kilpisjarvi (Figure 2). Tmax varied between 42.2 ± 1.0 in Svalbard and 57.8 + 9.3 at Toolik Lake Heath. Temperature range of growth, (Tmin - Tmax) varied between 48.7 and 65.2.

The MAST of soils was not significantly related with Topt (P= 0.5) nor was Tmin (P= 0.78, Adj. $R^2$ = --0.1). However, Topt did relate significantly with MaxT, increasing 0.63 °C per °C (Figure 1; P < 0.01, Adj. $R^2$ = 0.63). The temperature range of growth was significantly related to the amplitude of the temperature soil temperature (Linear regression; Adj. $R^2$= 0.3, P= <0.05).

### 3.2 Bacterial community composition

After filtering for singletons, we retrieved a total of 967,146 reads across the samples, belonging to 12692 ASVs. PERMANOVA analyses showed bacterial community composition to be significantly influenced by pH and MAST of the sampling sites (Table 2). The bacterial community composition was not significantly related with the Topt of the bacterial communities (P= 0.124). Proteobacteria (25.9%), Acidobacteriota (21.9%), Actinobacteriota (18.4%), Verrucomicrobiota (7%), Bacteroidota (6.7%), Chloroflexi (5.2%), Planctomycetota (5.1%), and Myxococcota (2.1%) were the most abundant phyla across all samples (Figure 3).

We observed only 12 ASVs that occurred at relative abundance greater than 0.001 in four or more sites (Table 3). Both regression tree and random forest analyses based on the relative abundance of these common ASVs showed the relative abundance of ASV11 was the best predictor of the corresponding community Topt (Suppl. Figure 1), in which it differentiated of ASV11 absence from the community and relative abundance > 0.055%. The pruned regression tree showed RMSE lower than the full tree on the validation dataset (Suppl. Figure 2; Suppl. Table 1). The linear regression model based on the MaxT as dependent variable showed larger predictive power of Topt than the pruned regression tree and random forest, since summarized across the 220 possible training sets, the median RMSE of for the linear model was lower than that median RMSE of the pruned tree and random forest (2.17, 4.14 and 3.51, respectively; Suppl. Figure 3).





## 4. Discussion

### 4.1 Temperature adaptation across the Arctic

In this study we have explored the role of soil thermal parameters on the temperature adaptation of soil bacterial communities in the Arctic. The cardinal points estimated from bacterial communities sampled at 12 Arctic locations were comparable to other bacterial communities from polar soils and showed a large variety between sites and soil types. We found $T_{min}$ to vary between -11.1 and -5.5 °C, which is comparable to soils sampled from sub arctic soils (Cruz-Paredes et al., 2021; Rinnan et al., 2011; Weedon et al., 2022). $T_{min}$ was lowest at the low arctic site Østerlien, which is lower than any the $T_{min}$ of previously

described for Arctic soil bacterial community, but fits within the range of $T_{min}$ of bacterial communities previously described in Antarctic soils (Rinnan et al., 2009). In contrast to $T_{min}$, $T_{opt}$ is hypothesized to vary less between thermal environments (Rinnan et al., 2009). At the Toolik Lake Moist Acidic Tundra site, estimated $T_{opt}$ was 23.5 °C, which is so far the lowest $T_{opt}$ described for a soil bacterial community in the Arctic (Rinnan et al., 2011; Weedon et al., 2022; Cruz-Paredes et al., 2021) and is also comparable to soil bacterial communities from Antarctica (Donhauser et al., 2020; Rinnan et al., 2009, 2011;

van Gestel et al., 2020). This site was characterized by relative low summer temperatures and moderate annual mean temperatures, compared to the other sites (Table 1).

### 4.2 Temperature adaptation is influenced by mean daily maximum soil temperature

Of the soil thermal parameters we tested, only MaxT had a significant correlation with temperature-growth relationships of Arctic bacterial communities. Temperatures above the optimum growth temperature can induce heat-related death of bacterial

cells, which results in a strong selective pressure by the maximum soil temperature on the bacterial community (Bárcenas-Moreno et al., 2009; Birgander et al., 2013; Donhauser et al., 2020). Consequently, the optimal growth temperature of soil bacterial communities is always observed to greatly exceed the maximum soil temperatures at a given location (Bárcenas-Moreno et al., 2009; Birgander et al., 2018; van Gestel et al., 2013; Rinnan et al., 2009). Our results show that even in cold biome environments the maximum soil temperature is an important determinant of the temperature physiology of soil bacterial

communities. While samples in this study were collected in summer, temperature-growth relationships are not affected by seasonal dynamics (van Gestel et al., 2013; Birgander et al., 2018), making it likely that the MaxT is the most important predictor of thermal adaptation amongst those we measured. All in all, the evidence collected in this study provides further support for the idea that temperature adaptation of soil microbial communities is best explained by the optimum-driven hypothesis (Alster et al., 2020). According to this hypothesis temperature-growth relationships are driven by the maximum

soil temperatures, and this was previously proposed as temperature adaptation could only be induced after exposure of communities to conditions above a certain threshold temperature (Bárcenas-Moreno et al., 2009; Birgander et al., 2013, 2018).



### 4.3 No evidence for influence of soil thermal parameters on Tmin or Tmax

In contrast to the clear relationship between MaxT and Topt, we found no evidence for a relationship between soil thermal parameters and the minimum and maximum cardinal points, nor with the thermal breadth of the bacterial temperature-growth relationships. This non significance could in principle be a result of statistical artefacts, since for the estimation of Tmin and Tmax, both cardinal points are extrapolated beyond the assay temperatures, which could cause a large standard deviation of the mean and increase the chance of type II errors. Indeed, the mean of site-level standard deviations across sites was relatively high for both Tmin and Tmax (respectively mean s.d. of 1.94 and 2.8). However, this variation was on the same order as that observed for Topt estimates amongst the sampled soil bacterial communities (mean s.d. of 2.06), implying that the lack of significance is most likely not due to limited power of the statistical analysis.

Given the importance of Tmin for determining activity at low temperatures, we expected that Tmin of communities would be related to site MinT. However, we did not detect a significant influence of MinT on the Tmin of soil bacterial communities. There is a general consensus that constantly frozen subsoils (permafrost) are an unlikely environment for proliferation of soil microbial life (Abramov et al., 2021). Due to this limited growth, cold-adapted (low Tmin) species might not necessarily thrive at subzero temperature but are likely to be better equipped to survive the winter conditions. Consequently, winter temperatures might not pose an environmental filter for the community assembly. Soil temperatures above freezing might have a larger influence on the temperature adaption of soil bacterial communities, when soil bacteria are most metabolically active (van Gestel et al., 2020). Therefore, the high soil temperatures in summer might induce a large environmental influence on the assembly of the bacterial communities. Additionally, strategies to survive subzero temperatures might not necessarily be indicative of the optimal growth temperature, as many microbial species that can cope with subzero temperature still grow best at relatively high temperature and are best described as psychro-tolerant rather than as true psychrophiles (Cavicchioli, 2015).These factors might therefore be the reason why we are unable to make predictions of Tmin based on the temperature parameters measured in this study.

Since MaxT influenced the Topt of the soil bacterial communities, we expected that this parameter would also correlate with the Tmax value of the soil bacterial community. Tmax has been hypothesized to increase with higher soil temperatures (Rinnan et al., 2009; Birgander et al., 2013), but to date this has not been directly tested. In our results, Tmax was not influenced by any of the measured soil thermal parameters. As noted above, Topt was far above maximum soil temperatures, which suggests that the measured growth rates of bacterial communities above Topt are rarely relevant in the soil environment. Therefore, it is likely that Tmax is less relevant for the performance of soil bacterial species and consequently, not subject to selection *in sensu* Vellend, 2010.

### 4.4 What will happen in response to warming?

Since MaxT was found to be most important predictor, it follows that changes to summer temperatures are likely to be the most important factor determining temperature-growth relationships of bacterial communities in Arctic soils under a changing



climate. Arctic summer air temperatures are predicted to increase less than the mean annual and winter temperature

(Karjalainen et al., 2018). While it has been estimated that mean annual soil temperature will rise ~ 2 - 4 °C around the Arctic by 2100 under RCP 4.5 (Aalto et al., 2018), accurate predictions of summer soil temperature in the Arctic are complicated by a variety of environmental factors that influence soil temperatures. At the local scale, soil temperatures are largely influenced by air temperature, solar radiation and precipitation (Karjalainen et al., 2018), leading to > 5 °C variation on the microscale (Aalto et al., 2013; Karjalainen et al., 2018). Increasing air temperatures in the Arctic can also lead to changes in vegetation

height and shrub expansion (Mekonnen et al., 2021), which moderate increasing soil temperature by shading during the summer season (Paradis et al., 2016; Blok et al., 2011). Furthermore, it is likely that the Arctic terrestrial region will experience more frequent and extreme heatwaves, which could induce rapid change in the temperature-growth relationships if soil temperatures exceed historical maximum soil temperatures and/or the Topt of the soil bacterial communities (Bárcenas-Moreno et al., 2009; Birgander et al., 2013; Donhauser et al., 2020). These complicated local scale effects imply that more

microclimatic data will be needed for more accurate assessments of temperature adaptation of soil bacterial communities in the Arctic.

We computed the optimal growth temperature of soil bacterial communities across the Arctic based on combining the Soil Temp database (Lembrechts et al., 2021) with our estimates of the relationship between soil temperature (MaxT) and Topt (Figure 4). Our study covered a large portion of the range of maximum soil temperature within the Arctic region, as these

temperatures currently vary between -0.4 and 20.6 °C (Lembrechts et al., 2021). Figure 4 shows that the Topt of Arctic soil bacterial communities likely varies between 22 and 35 °C. A combination of this pan-arctic projection, predicted future summer (soil) temperatures, and other spatial databases such as soil C maps, could be useful to identify locations where soil bacterial communities will be sensitive to future warming, where potential shifts in the temperature-growth relations can occur, and where these may have disproportionate impacts on regional biogeochemistry. For example, by identifying regions where

local soil temperatures are expected to rise rapidly and soil organic stocks are large.

### 4.5 Can we use microbial community data for predicting temperature adaption?

Predicting temperature adaptation of soil bacterial communities across the Arctic might be limited by lack of long term soil temperature data across the Arctic as most Arctic research has focused on only few research sites (Metcalfe et al., 2018). To

explore the potential use of microbial 'bio-indicators' for predicting the temperature-growth relationships of *in situ* soil bacterial communities (Hicks et al., 2021), we evaluated whether microbial community data can reveal the temperature adaptation of microbial communities. We found that regression tree analysis using bacterial ASVs as potential predictors (Suppl. Fig. 4) produced larger estimation errors on prediction of the Topt of soil bacterial communities when compared with the linear regression against MaxT. This can be partially attributed to the low effective sample sizes resulting from the use of

cross-validation methods to prune the regression trees, but likely also reflects a lack of consistent signal in the bacterial composition data. Although this doesn't refute the potential for using compositional data to predict community-broad



temperature growth relationships (Hicks et al., 2021), it implies that such methods would need a larger training dataset with more sample sites for proper validation, and more accurate predictions. The full regression tree used a low number of ASVs (Supplementary Figure 2, Supplementary Table 2), which were not observed in all soil types, which might indicate limited use for other datasets. This suggests that indicator species, if they exist, might be indicative of the temperature adaptation of bacterial communities for only certain particular soil types or climatic regions. Despite these caveats, it is notable that the pruned regression tree and random forest model both identified the abundance of ASV11 as effective in predicting the Topt (Supplementary Figure 1). ASV11 matches 100% to ASV that is the most commonly observed bacterial in arctic soils (Malard et al., 2019). The genus of Candidatus Udaeobacter, to which ASV11 matches, is commonly found in soil environments globally (Brewer et al., 2016). It has been proposed to be a small oligotrophic and resilient soil bacteria characterized by aerobic heterotrophic metabolism with small genome size (2.8-3.2 Mbp), large diversity of antibiotic resistance genes and a preference for acidic soils (Brewer et al., 2016; Willms et al., 2020, 2021). However, so far no study has successfully cultivated the any lineage of the genus 'Candidatus Udaeobacter' and traits related to temperature preferences have not been recorded. In the pruned tree (Supplementary Figure 4) the presence or absence of ASV11 was indicative a Topt of 26.3 or 31.4, respectively. As this taxon was absent in 7 out of 12 soils, the utility as an indicator of temperature adaption is quite limited. In summary, although there is some potential utility in using community data to estimate and predict aspects of soil bacterial temperature physiology, our results suggest that more accurate predictions can be made from soil temperature records.

## 5. Conclusions

In this study, we showed a large variety in the temperature adaptation of soil bacterial communities from the sub to High Arctic region. Due to the large influence of maximum soil temperatures, we predict that summer warming, to the extent that leads to higher maximum soil temperatures, will lead to increasing community-level increase the Topt of these bacterial communities under future climate conditions in the Arctic. The influence of shifting optimal growth temperature for soil bacterial communities on soil carbon cycling will need further investigation to evaluate the contribution to the vulnerability of soil carbon stock in the Artic under future climate conditions.





**Appendices**

**Supplementary Figure 1**. Importance of variables for Random Forest Tree Regression Analysis

**Supplementary Figure 2**. Regression tree analysis for estimation of Topt of bacterial community based on the relative
abundance of the 12 common ASVS across all soil samples.

**Supplementary Figure 3.** Histograms of the performance of each model type by RMSE.

**Supplementary Figure 4.** Scatterplot the relationship between Topt of the soil bacterial community and relative abundance
of ASV 11 in the community composition. Lines indicate the predicted Topt values by the pruned regression tree indicate.

**Supplementary Table 1.** Characteristics of the soil types

**Supplementary Table 2.** Taxonomy of ASVs commonly found in samples in study and previous work. N= indicates the
number of soil types the ASV was present in this study.

**Data availability**

The data used for statistical analysis and visualization is available on Figshare (doi; 10.6084/m9.figshare.20977588, doi:
10.6084/m9.figshare.20977582, doi: 10.6084/m9.figshare.20977585). The raw sequences of the 16S rRNA gene amplicon
sequences have been deposited on NCBI's Sequence Read Archive under BioProject PRJNA856638.

**Authors Contributions**

RR and JTW designed the study. MD and RR performed soil analysis and bacterial community analysis. RR performed leucine
assays and statistical analysis. All authors contributed to writing of the manuscript.

**Competing interests**

The authors declare that they have no conflict of interest.

**Acknowledgements**

The research leading to these results has received funding from the European Union's Horizon 2020 project INTERACT,
under grant agreement No 730938. RR and JTW were supported by NWO Netherlands Polar Program (project id 866.16.042).

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





**Figure Captions**

**Figure 1**. A) Map with polar projection showing the 12 sampling sites across the Arctic and B) the average values of soil thermal regimes for each site including the maximum (red), mean (grey) and minimum (blue) soil temperature. Error bars indicate the standard deviation across the years.

**Figure 2**. A) Estimated growth curves for each soil type depicted by the normalized leucine incorporation over incubation temperature. Colors indicate the maximum soil temperature of each sampling site. B) Linear relationships between the optimal 565 growth temperature and maximum soil temperature, error bars indicate the standard error.

**Figure 3**. Bar plot showing the relative abundance (%) of top 10 most abundance phyla across all soil samples. Color shades indicate the two most abundant Order for each of these phyla.

**Figure 4**. Map of the predicted Topt of soil bacterial communities across the Arctic based on the linear relationship between 570 maximum soil temperature (from SoilTemp database) and Topt.

**Table 1**. Thermal regimes of the 12 soil types

**Table 2**. Results of PERMANOVA showing the influence of soil parameters on the bacterial community composition


**Table 3.** Taxonomy of commonly observed bacterial ASVS. N indicates number of soils the ASV was observed in.





**Table 1. Thermal regimes of the 12 sampling sites in ° C.** MaxT depicts the warmest day of the year, MAST the mean
annual temperature and Min the coldest day of the year. ± indicate standard deviation of the mean value recorded of the
temperature record from the first year (Start) till the last year (End). Depth indicated the soil temperature logger depth in
centimeters.

| Site | Start | End | Depth | MaxT | MAST | MinT |
|---|---|---|---|---|---|---|
| **Abisko** | 2015 | 2019 | 10 | 13.2 ± 3.7 | 0.8 ± 0.1 | -9.8 ± 5.3 |
| **Blaesedalen** | 2013 | 2020 | 10 | 10.5 ± 1.3 | -0.7 ± 1.3 | -14.5 ± 4.8 |
| **Iceland Forest** | 2013 | 2019 | 10 | 11.8 ± 0.3 | 5.1 ± 0.3 | 0 ± 0.3 |
| **Iceland Grassland** | 2013 | 2018 | 10 | 14.2 ± 0.7 | 6.1 ± 0.6 | 0.1 ± 0.2 |
| **Inuvik** | 2002 | 2018 | 5 | 13.5 ± 2.2 | 1.3 ± 0.7 | -6.1 ± 3 |
| **Kobbefjord** | 2008 | 2019 | 10 | 12.3 ± 0.8 | 1.9 ± 0.5 | -5.3 ± 2.5 |
| **Svalbard** | 2008 | 2016 | 25 | 9 ± 0.8 | -3.5 ± 0.5 | -18.5 ± 2.3 |
| **Svanhovd** | 2014 | 2021 | 10 | 15 ± 0.7 | 3.6 ± 0.6 | -3.4 ± 0.1 |
| **Toolik Lake Heath** | 2002 | 2019 | 10 | 13.5 ± 1 | -2.2 ± 1.2 | -13.8 ± 3.8 |
| **Toolik Lake MAT** | 2008 | 2021 | 10 | 2.7 ± 0.4 | -2.2 ± 0.6 | -8.1 ± 2.2 |
| **Toolik Lake MNAT** | 2012 | 2021 | 10 | 4.7 ± 0.8 | -1.7 ± 0.8 | -8.2 ± 2.9 |
| **Kilpisjarvi** | 2019 | 2019 | 10 | 16.3 | | |





**Table 2. Results of PERMANOVA on bacterial community composition**

|  | Df | SumOfSqs | R2 | F | Pr(> F) |
|---|---|---|---|---|---|
| MAST | 1 | 0.013 | 0.061 | 3.136 | **0.037** |
| pH | 1 | 0.085 | 0.412 | 21.115 | **0.001** |
| Water Content | 1 | 0.008 | 0.037 | 1.882 | 0.115 |
| Topt | 1 | 0.008 | 0.037 | 1.919 | 0.124 |
| Organic C | 1 | 0.004 | 0.021 | 1.084 | 0.290 |
| Organic N | 1 | 0.008 | 0.040 | 2.068 | 0.095 |
| Residual | 20 | 0.081 | 0.391 |  |  |
| Total | 26 | 0.207 | 1 |  |  |




**Table 3. Taxonomy of commonly observed bacterial ASVS. N indicates number of soils the ASV was observed in.**

| OTU | PHYLUM | CLASS | ORDER | FAMILY | GENUS | SPECIES | N |
|---|---|---|---|---|---|---|---|
| 1 | Acidobacteriota | Acidobacteriae | Acidobacteriales | Acidobacteriaceae (Subgroup 1) | Granulicella | Uncultured soil | 4 |
| 2 | Acidobacteriota | Acidobacteriae | Subgroup 2 | Subgroup 2 | Subgroup 2 | Uncultured forest | 4 |
| 3 | Acidobacteriota | Acidobacteriae | Subgroup 2 | Subgroup 2 | Subgroup 2 | Uncultured eubacterium | 6 |
| 4 | Actinobacteriota | Thermoleophilia | Solirubrobacterales | Solirubrobacteraceae | Conexibacter | | 4 |
| 5 | Actinobacteriota | Actinobacteria | Corynebacteriales | Mycobacteriaceae | Mycobacterium | | 4 |
| 6 | Actinobacteriota | Actinobacteria | Frankiales | Acidothermaceae | Acidothermus | | 4 |
| 7 | Proteobacteria | Alphaproteobacteria | Rhizobiales | Xanthobacteraceae | | | 10 |
| 8 | Proteobacteria | Alphaproteobacteria | Rhizobiales | Xanthobacteraceae | Bradyrhizobium | | 4 |
| 9 | Proteobacteria | Alphaproteobacteria | Rhizobiales | Xanthobacteraceae | | | 5 |
| 10 | Proteobacteria | Alphaproteobacteria | Rhizobiales | Xanthobacteraceae | | | 7 |
| 11 | Verrucomicrobiota | Verrucomicrobiae | Chthoniobacterales | Chthoniobacteraceae | Candidatus Udaeobacter | Uncultured soil | 5 |
| 12 | Proteobacteria | Gammaproteobacteria | WD260 | WD260 | WD260 | Uncultured eubacterium | 4 |





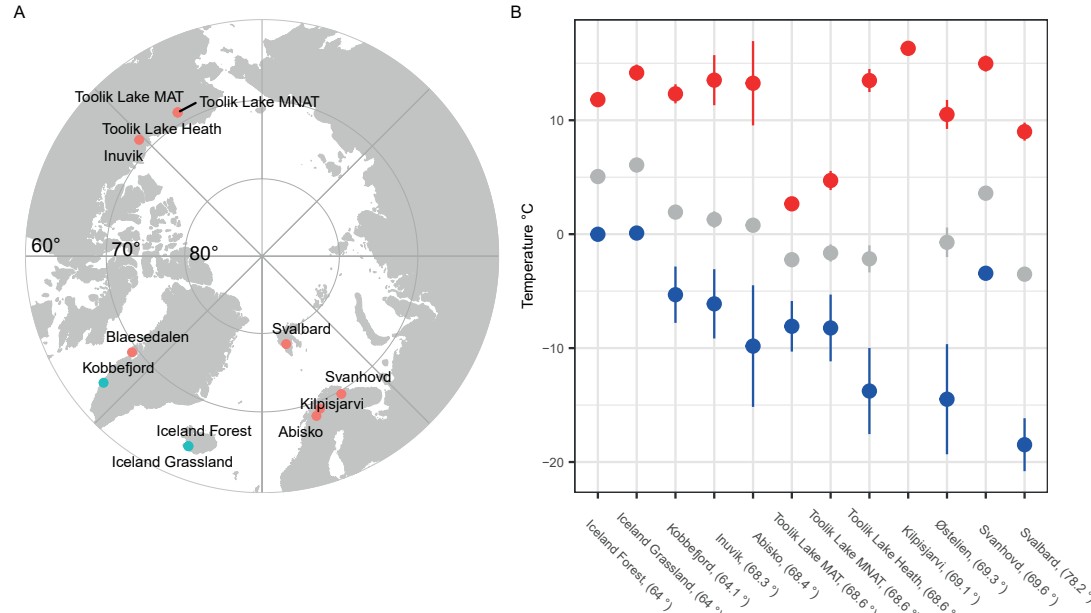

**Figure 1**. A) Map with polar projection showing the 12 sampling sites across the Arctic and B) the average values of soil thermal regimes for each site including the maximum (red), mean (grey) and minimum (blue) soil temperature. Error bars indicate the standard deviation across the years.



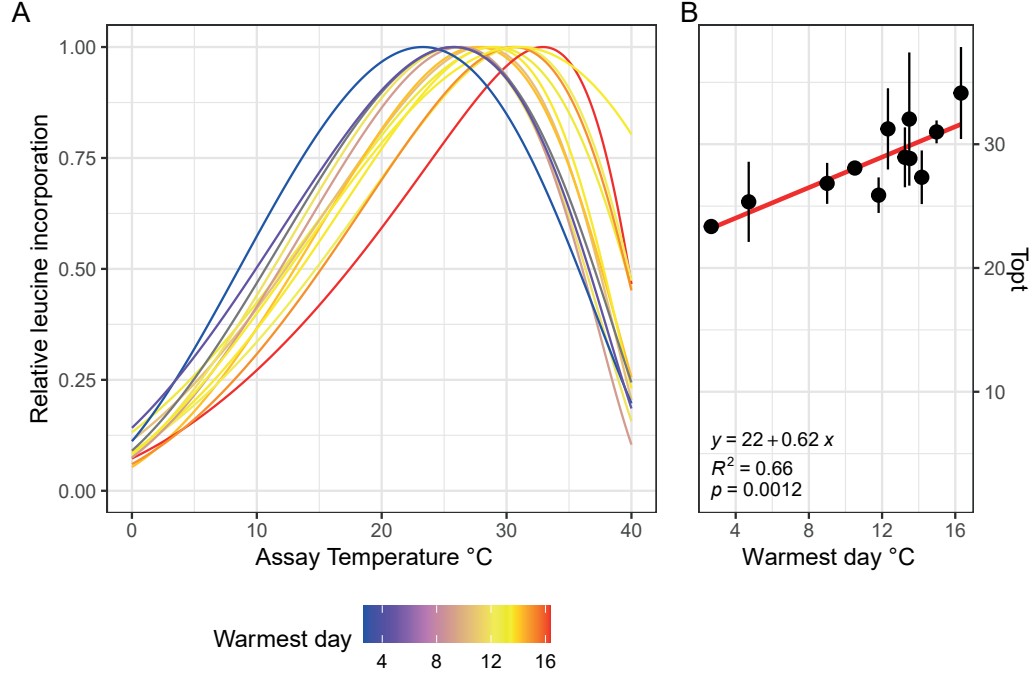

**Figure 2**. A) Estimated growth curves for each soil type depicted by the normalized leucine incorporation over incubation temperature. Colors indicate the maximum soil temperature of each sampling site. B) Linear relationships between the optimal growth temperature and maximum soil temperature, error bars indicate the standard error.





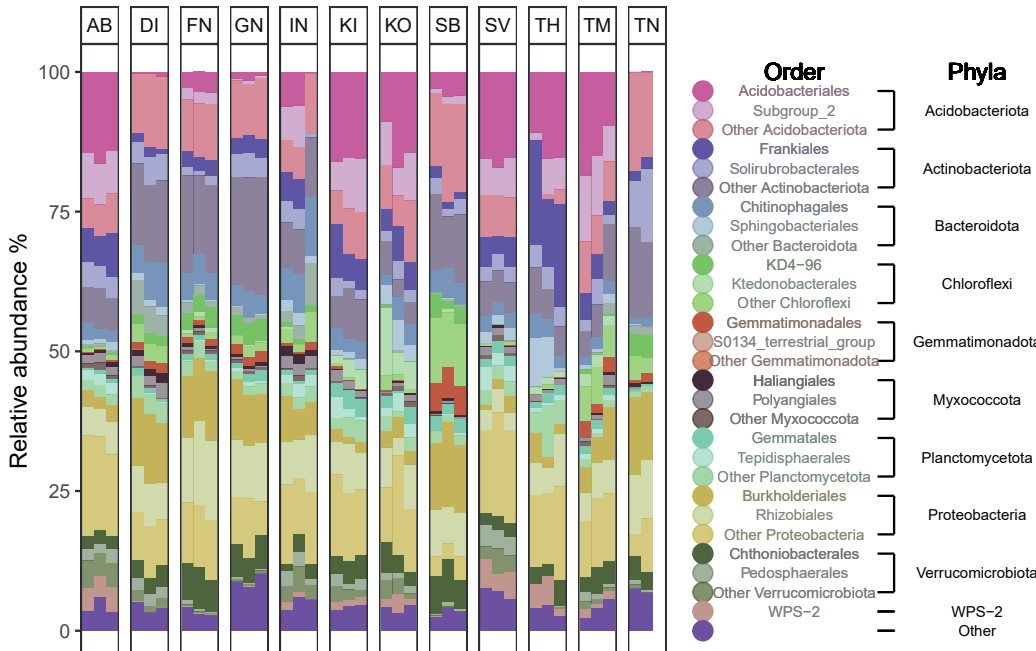

**Figure 3**. Bar plot showing the relative abundance (%) of top 10 most abundance phyla across all soil samples. Color shades indicate the two most abundant Order for each of these phyla.

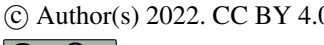



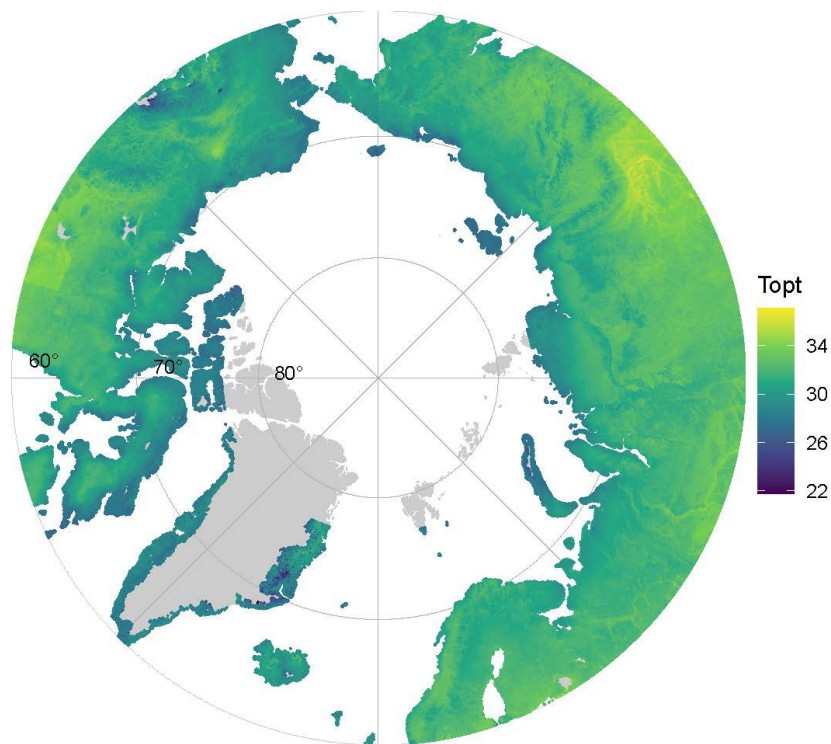

**Figure 4**. Map of the predicted Topt of soil bacterial communities across the Arctic based on the linear relationship between maximum soil temperature (from SoilTemp database) and Topt.