# Peer review of "Maximum summer temperatures predict the temperature adaptation of Arctic soil bacterial communities"

_Biogeosciences, 2022_

## Author Response (AR1)

We thank the reviewers for their thoughtful feedback and address the changes that were made to the manuscript in correspondence to their comments. Line numbers refer to the changes in the revised manuscript with track-changes.

**1) The authors use both OTU and ASV in the manuscript. The pipeline in the methods result in ASVs, so perhaps best to stick to ASV for consistency.**

-- In the introduction we now refer to taxa instead of OTUs when referring to previous studies that used OTUs (Line 95 & 99). Due to a typo in Table 4 we referred to OTUs, which will be changed to ASV.

**2) p.7, line 170 - put parentheses around 2012, and remove it from Caporaso.**

-- This was be changed to Caporaso et al., (2012).

**3) p.10 line 255 remove "have"**

-- We removed "have" from line 255.

**4) p.11: line 302, add space before "These". Line 310, add parentheses around "2010", and line 312: add "the" to "most important predictor"**

-- We corrected these typos.

**5) p.12: line 346 - do not contract words. Spel out "doesn't" to "does not"**

-- This was changed.

Thank you for the thoughtful feedback. Below we answer the questions and show how we like to implement these concerns.

*- Line 130-135: how was the soil temperature recorded on the sites exactly? Table 3 reports on temperature recorder at different depth. How these different in depths (from 5 to 25 cm) were accounted in the analysis?*

For all sites, there was soil temperatures available that was recorded from permanent temperature loggers. The difference in depth between the sites results from a limitation in data availability. We did not directly account for the difference in the depth of the soil loggers for Inuvik and Svalbard sites during the statistical analysis. For three other sites where more depth resolution was available we initially saw a deviation of <0.5 °C for the mean annual soil temperature over this depth range.

As the amplitude of temperature dampens soil depth, the difference in the extreme temperature is more pronounced. For the mean annual maximum soil temperature there was from 5 to 10 cm a difference +0.5°C deviation (n=2) and 10-20 cm depth a deviation of -1.4 °C (n=2). For the minimum soil temperature we saw a deviation of -0.55°C between 5 to 10 cm (n=2), and a deviation of +0.96 °C from 10 to 20 cm (n=2).

This suggests that the error in soil temperature parameters arising from a mismatch between sampling depth and temperature logger depth would be on the order of <0.5, 1.5 , an 1 °C for parameters MAT, MaxT, MinT respectively. This is equivalent to 2, 10 and 5% of the total gradient length in our study.

To test the robustness of our conclusions to these errors, we re-ran the analyses excluding Inuvik and Svalbard from the linear regressions. The resulting models were showed equivalent to the models using the full dataset, showing only a significant correlation between Topt and maximum soil temperature. The revised version of this model, showed slightly altered coefficient (from 0.6286 to 0.6244) and intercept (from 21.3386 to 21.6027).

We added this extra analysis in the supplementary methods and discuss the outcomes briefly in the discussion (Line 298-290)

*- Were other temperature predictor considered (such as temperature variation, number of days in minus temperature, etc.) which would be relevant for soil microbial communities. The choice of only 3 temperature predictors seems a bit limited.*

We support the idea to test more possible predictors. We had already tested the total amplitude of the soil temperature (line 239-240). We also added test for days > 0°C and mean temperature for winter and summer (Line 236-239).

*- Line 160: what does it mean for 24 – 2 hours?*

The incubation time depended on the assay temperature, where it was 24 h at 0°C, 8 h at 4°C, 4 h at 10 and 15°C and 2 h at 24.5, 28.5, 33 and 40°C. We have added the specific incubation time for each assay temperature to the Methods section on Line 160.

*- Amplicons sequences were analyzed as ASVs but OTUs are reported in table 3.*

Thank you noticing, this contains a typo and was changed to ASV (Table 4).

*- Line 200: The sentence "tested the relationship between Tmin and minimum soil temperature, Tmax and the maximum soil temperature…" is confusing. Also, Tmax/MaxT and Tmin/minT: these abbreviations are somewhat confusing to the reader through the manuscript as they designed two different things. Especially during the discussion, it is sometimes hard to follow (lines 283-310). Please clarify the parameters used (maybe in a form of a small table/list) and the related abbreviations.*

We changed abbreviations for soil temperature and italicized them for a more clear distinction between the terms referring to temperature-growth relationships and soil temperature throughout the manuscript and added table 2 for an overview.

*- Lines 237-239: why MaxT, MinT were not part of the PERMANOVA analyses presented in Table 2? I think MaxT would especially be important to test its correlation to the total microbial community composition as this parameter revealed to be important for microbial growth. I also think that adding a figure showing the correlation of the predictors to the whole community structure would be meaningful (such as an ordination plot with significantly correlated variables).*

Due collinearity between the MAT, MaxT and MinT, it would not be appropriate to include all predictors in the PERMANOVA. Therefore, we have added two seperate PERMANOVA models using either *STmax* or *STmin* (Supl. Table 2 & 3). We also added a PCOA plot with all variables (Figure 5).

*- Lines 243-251: I find a bit reductive to work only with 12 selected ASVs to test the microbial adaptation on temperature. I wonder why these selection criteria (>0.001% in at least 4 soils) were chosen? Could it be possible to use less restrictive criteria to be able to test for the link between community composition and temperature more largely. Would the output of the regression tree and the random forest analyses be similar then?*

There were no taxa present in all soils, which off course would be even more useful. Therefore, we tried to choose a balance between commonly observed and larger numbers species from the table below.

| | | | | | Abunce % | | | |
|---|---|---|---|---|---|---|---|---|
| | | *0.1* | *0.02* | *0.01* | *0.004* | *0.002* | *0.001* | *5.00* |
| | *2* | 89 | 117 | 118 | 118 | 118 | 118 | 11 |
| | *3* | 22 | 31 | 32 | 32 | 32 | 32 | 3 |
| | *5* | 9 | 11 | 12 | 12 | 12 | **12** | 1 |
| | *6* | 5 | 5 | 5 | 5 | 5 | 5 | 5 |
| occur- | *7* | 1 | 3 | 3 | 3 | 3 | 3 | 3 |

| enc (n sites) | | | | | | | | |
|---|---|---|---|---|---|---|---|---|
| | **9** | 1 | 1 | 1 | 1 | 1 | 1 | 1 |
| | **12** | 0 | 0 | 0 | 0 | 0 | 0 | 0 |

Initial attempts also included e.g. ASVs shared between 3 soils. However, the random forest analysis and pruning showed that the relative importance was larger for the subset of more commonly observed 12 ASVs than the other ASVs observed in 3 soils. We now discuss the filter cut-offs and resulting number of ASVs on line 252-256.

To test whether less restrictive cut-offs lead to different results we have added regression trees and random forest models for these different datasets of common ASVs (common in only 2 or 3 soils; Supplementary Table 4), which led to similar results (Line 254-256).

*- Lines 268-269: The significance of MaxT in the temperature-growth relationship is one of the main output of the study. However, I can't find*

The significance is described on line 236 and Figure 2. We will added a reference to Figure 2 on line 282.

*- Lines 327 – 331: This sentences should be part of the results section.*

We moved these sentences to the first paragraph of the results, insert at line 241-243.